# The self-organization of grid cells in 3D

**Federico Stella\*, Alessandro Treves**

Cognitive Neuroscience, Scuola Internazionale Superiore di Studi Avanzati, Trieste, Italy

**Abstract** Do we expect periodic grid cells to emerge in bats, or perhaps dolphins, exploring a three-dimensional environment? How long will it take? Our self-organizing model, based on ring-rate adaptation, points at a complex answer. The mathematical analysis leads to asymptotic states resembling face centered cubic (FCC) and hexagonal close packed (HCP) crystal structures, which are calculated to be very close to each other in terms of cost function. The simulation of the full model, however, shows that the approach to such asymptotic states involves several sub-processes over distinct time scales. The smoothing of the initially irregular multiple fields of individual units and their arrangement into hexagonal grids over certain best planes are observed to occur relatively quickly, even in large 3D volumes. The correct mutual orientation of the planes, though, and the coordinated arrangement of different units, take a longer time, with the network showing no sign of convergence towards either a pure FCC or HCP ordering.

## Introduction

Where does our internal representation of space come from? And how does it code for space extending in three dimensions? New findings about space-related activity in the bat have recently raised this question again (*Ulanovsky and Moss, 2011*; *Yartsev et al., 2011*; *Yartsev and Ulanovsky, 2013*; *Finkelstein et al., 2014*). The similarity in the place cell and, most remarkably, in the grid cell representation between rodents and bats suggests a common neural substrate for spatial navigation, shared across these mammals (*Andersen and Buneo, 2002*; *Jacobs et al., 2010*; *Sereno and Lehky, 2011*; *Killian et al., 2012*; *Indovina et al., 2013*; *Jacobs et al., 2013*; *Thurley et al., 2014*), and it provides an indication possibly valid also for other animals living and moving extensively in three dimensions, like for example dolphins, monkeys and even non-mammalian species (*Healy et al., 2005*; *Dacke and Srinivasan, 2007*; *Wu and Dickman, 2012*; *Burt de Perera and Holbrook, 2012*). At the same time, the obvious difference in the behavior of these species requires a system that flexibly adapts to perform computations as different as mapping two- or three-dimensional space (*Knierim et al., 2000*; *Hayman et al., 2011*; *Taube and Shinder, 2013*). Here we describe a model of grid cell formation that accounts for both these aspects of spatial cognition, in a unitary perspective on the mEC network (*Figure 1*).

Grid cells seemingly require some clever engineering design that generates the common periodicity among neighboring units while keeping them distinct in terms of spatial phase (*Zilli, 2012*). While place cell and head direction cells have been shown to directly generalize to three dimensions (*Yartsev and Ulanovsky, 2013*; *Finkelstein et al., 2014*), the form that grid cells will exhibit in higher dimensionality (currently tested in flying bats [*Ginosar et al., 2014*]) is still not clear. Further, the question is still open of how the mechanism producing such a complex periodic pattern on a surface can, in the case of bats, extend to a volume (*Jeffery et al., 2013*), even when accepting the information-theoretic optimality of a regular lattice (*Mathis et al., 2014*). In the self-organization perspective that we propose, the spatial responses first emerge spontaneously, at the single unit level, with no engineering required. In the simplest version of the model, which we have explored in a series of studies (*Kropff and Treves, 2008*; *Si et al., 2012*; *Stella et al., 2013*), the periodicity of the

\*For correspondence: fstella@sissa.it

**Competing interests:** The authors declare that no competing interests exist.

**eLife digest** Our ability to navigate through our environment depends on a region of the brain called the hippocampus. In the 1990s it was shown that this structure, which takes its name from the Greek word for 'seahorse' owing to its shape, was larger in London taxi drivers than it was in the general population. However, as early as the 1960s, experiments in rats had revealed that specific cells within the hippocampus—called place cells—fire whenever an animal is in a particular location, and thus enable the animal to build up a map of its environment.

In 2014, the scientist who discovered place cells shared the Nobel Prize in Physiology or Medicine with two neuroscientists who had discovered an additional type of cell that is involved in navigation. These grid cells, which are located in a region of the brain that provides input to the hippocampus, 'fire' at multiple points in space. When the scientists who discovered grid cells plotted these points in two dimensions, they formed a grid of tessellating triangles that spanned the entire area.

However, many animals, including aquatic mammals, monkeys and bats, navigate in three dimensions rather than two. This raises an obvious question: can grid cells also represent three-dimensional space? Stella and Treves have addressed this issue by constructing a computer model that simulates grid cell activity in a virtual bat flying through a virtual room. The model reveals that grid cells switch from firing largely at random to firing in some semblance of a three-dimensional pattern relatively quickly.

However, this pattern bears little resemblance to the highly ordered arrangement seen in two dimensions. Indeed, the model suggests that a bat flying at 1 metre per second around a room that measured $2.5 \times 2.5 \times 2.5$ metres would need to fly continuously for a very long time (at least 80 hours) before such a pattern could be established in three dimensions. This suggests that the regular tessellation shown by grid cells in two dimensions might not be routinely established in three dimensions. Instead, simpler 'precursor' firing patterns may form over shorter periods of time, providing a looser mapping of three-dimensional space.

grid pattern is a result of firing rate adaptation during exploration sessions that span a considerable developmental time (*Figure 1, c*). It is fixated gradually by means of synaptic plasticity in the feed-forward connections, which convey broad spatial inputs, for example but not necessarily from 'place units' (*Figure 1, b*). Contrary to other models limited to the explanation of grid cell expression, this model delves into the issue of their induction and, most importantly, can account for the effects of the geometry of the explored environment on the outcome of the self-organization process. We have shown how, for example, the model produces pentagonal grids on a spherical surface and heptagonal ones on a hyperbolic one (*Urdapilleta et al., 2015*). The nature of our model allows us to now investigate the process of grid cell self-organization in three dimensions without the need to modify any of its features. We use bats as our reference, as it is the species currently available for experiments during roughly homogeneous navigation along the three dimensions of physical space (*Figure 1, a*).

## Model

### Numerical simulations

In our simulations a virtual bat explores a volume of side $L$ with a constant speed $v$. The position of the animal is sampled at time steps of constant $\Delta t$. We temporarily leave these quantities unspecified. We will discuss their actual values at the end of the paper as they are critical for the interpretation of the results. For the moment they should be understood as expressed in arbitrary units. The path the animal performs is generated as a correlated random walk in which the direction of movement at any time step depends on the previous one. For simplicity, the change in running direction between two consecutive steps of the virtual bat is sampled from a Gaussian distribution with zero mean and standard deviation $\sigma_h = 0.15$ radians.

### The network

Our model is comprised of two layers. The input network represents, for example, the CA1 region of the hippocampus and contains $N_{inp} = 12^3$ units. The output network represents a population of

$N_{mEC} = 125$ would-be grid units in mEC, all with the same adaptation parameters when not differently stated. Each mEC unit receives afferent spatial inputs which, as already discussed in *Kropff and Treves (2008)*, we take for simplicity to arise from regularly arranged place cells, although they could also arise from spatially modulated units in the adjacent cortices. The input to unit $i$ at time $t$ is then given by $h_i^t$:

$$h_t^i = \sum_j W_{ij}^{t-1} r_j^t. \tag{1}$$

The weight $W_{ij}^t$ connects input unit $j$ to mEC unit $i$. We will assume that at the time the mEC units develop their firing maps, spatially modulated or place cell-like activity is already present, either in the parahippocampal cortex or in the hippocampus. The network model works in the same way with any kind of spatially modulated input, but the place cell assumption reduces the averaging necessary for learning. Moreover, as recent studies have shown (*Langston et al., 2010*; *Wills et al., 2010*), it is entirely plausible that place cells develop an adult-like spatial code earlier than grid cells do. We thus model the place field as a Gaussian bump centered in the place cell preferred position $\vec{x}_{j0}$:

$$r_j^t = \exp\left(-\frac{\|\vec{x}^t - \vec{x}_{j0}\|^2}{2\sigma_p^2}\right), \tag{2}$$

where $\vec{x}^t$ is the position at time $t$ of the simulated bat, $\sigma_p = 0.05\,L$ is the width of the firing field and $\|a - b\|$ is the Euclidean distance between two points $a$ and $b$ in three dimensions. Place field centers

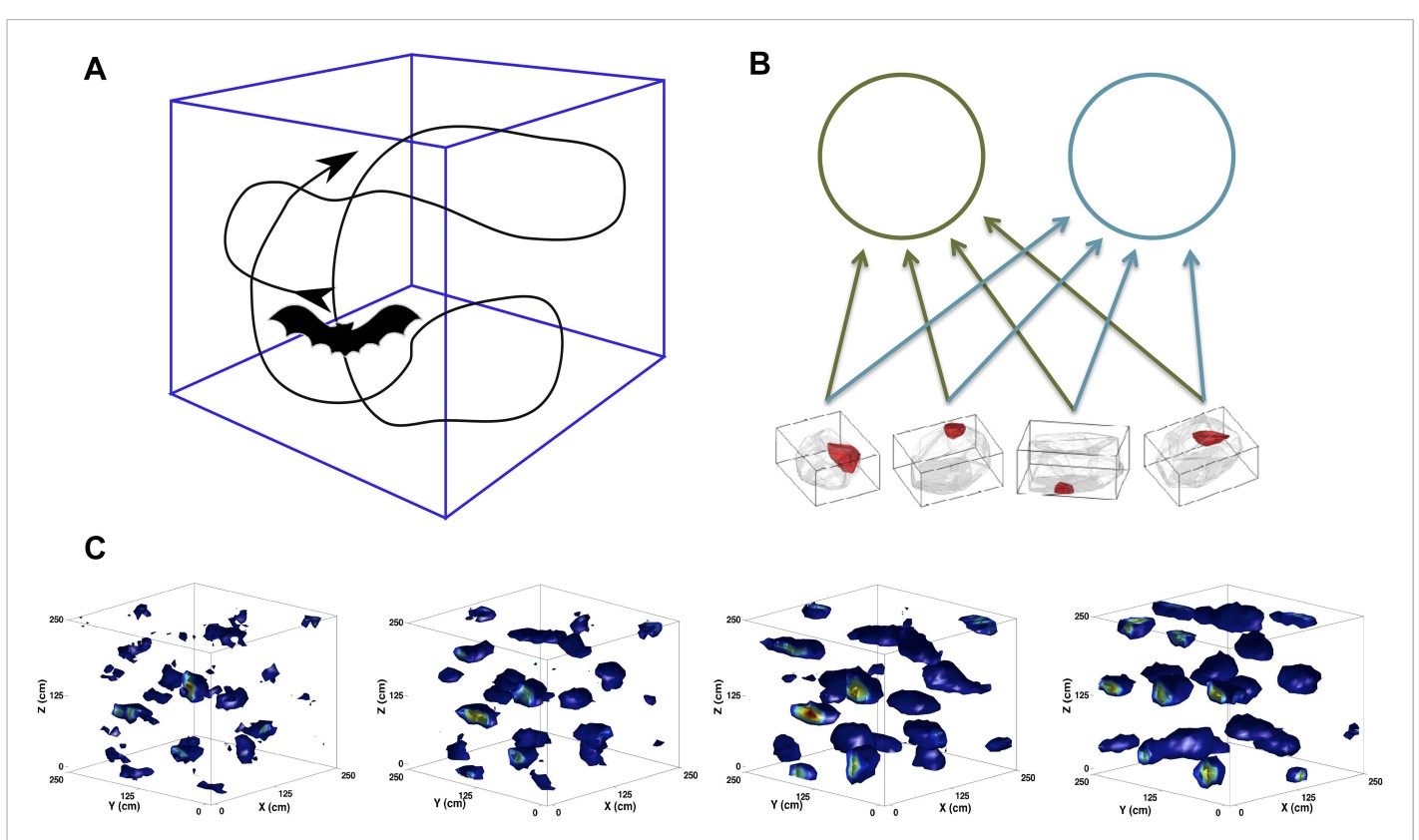

**Figure 1**. Scheme of the simulations. (**A**) We simulate the trajectory of a bat exploring a volume of space over a prolonged period of time. At each step, the bat moves forward at a constant speed and chooses a new direction of movement close to the previous one. (**B**) The feed-forward network (equivalent to that shown in *[Kropff and Treves, 2008]*), including here only two mEC neurons. The would-be grid units receive inputs from place cell-like units with firing fields similar to those reported in *Yartsev and Ulanovsky (2013)*. (**C**) Snapshots of the evolution of the firing rate map of a single unit. The figures correspond to (from left to right) 2, 5, 10 and 20 million time steps of learning.

© 2013, M Yartsev, N Ulanovsky. Figure 1 is reproduced from M Yartsev, N Ulanovsky. 2013. Representation of three-dimensional space in the hippocampus of flying bats. *Science* **340**:367–372. Reprinted with permission from AAAS.

are homogeneously distributed in the volume (consistently with the experimental data presented in [Yartsev and Ulanovsky, 2013]). Note that the choice of the value for the parameter $\sigma_p$ is not itself crucial for our model (Kropff and Treves, 2008). The range of values that it can take in simulations, however, is constrained in practice by the number of place cells used in the model. Basically the dimension of each place field should guarantee a homogeneous amount of global input activity across the environment. Given a sufficient place field density in the input layer, the parameter does not play a critical role anymore. As we show in the following, the properties of the developing grid fields depend on the time scale of adaptation and not on the size of the place fields.

## Single unit dynamics

The firing rate $\Psi_i^t$ of mEC unit $i$ is determined by a non-linear transfer function

$$\Psi_i^t = \frac{2}{\pi} \arctan\left[g^t\left(\alpha_i^t - \mu^t\right)\right]\Theta\left(\alpha_i^t - \mu^t\right), \tag{3}$$

which is normalized to have maximal firing rate equal to 1 (in arbitrary units), and $\Theta(\cdot)$ is the Heaviside function. The variable $\mu^t$ is a threshold while $\alpha_i^t$ represents the adaptation-mediated input to the unit $i$. It is related to $h_i^t$ as follows:

$$\begin{aligned}
\alpha_i^t &= \alpha_i^{t-1} + b_1\left(h_i^{t-1} - \beta_i^{t-1} - \alpha_i^{t-1}\right), \\
\beta_i^t &= \beta_i^{t-1} + b_2\left(h_i^{t-1} - \beta_i^{t-1}\right),
\end{aligned} \tag{4}$$

where $\beta_i$ has slower dynamics than $\alpha_i$, with $b_2 = b_1/3$, $b_1 = 0.1$ (in a continuous formulation, the $b$ coefficients become rates, in units of $(\Delta t^{-1})$). This adaptive dynamics makes it more difficult for a neuron to fire for prolonged periods of time, and corresponds to the kernel $K$ considered in the analytical treatment (Kropff and Treves, 2008). The gain $g^t$ and threshold $\mu^t$ are iteratively adjusted by Equation 5 at every time step to fix the mean activity $a = \sum_i \Psi_i^t / N_{mEC}$ and the sparsity $s = \left(\sum_i \Psi_i^t\right)^2/\left(N_{mEC} \sum_i \Psi_i^{t2}\right)$ within a 10% relative error bound from pre-specified values, $a_0 = 0.1$ and $s_0 = 0.3$, respectively. If $k$ is indexing the iteration process:

$$\begin{aligned}
\mu^{t,k+1} &= \mu^{t,k} + b_3\left(a^k - a_0\right), \\
g^{t,k+1} &= g^{t,k} + b_4 g^{t,k}\left(s^k - s_0\right).
\end{aligned} \tag{5}$$

$b_3 = 0.01$ and $b_4 = 0.1$ are also rates, but in terms of intermediate iteration steps. $a^k$ and $s^k$ are the values of mean activity and sparsity determined by $\mu^{t,k}$ and $g^{t,k}$ in the intermediate iteration steps. The iteration stops once the gain and threshold have been brought within the 10% error range, and the activity of mEC units is determined by the final values of the gain and threshold in Equation 3.

## Synaptic plasticity model

The learning process modifies the strength of the feed-forward connections according to a Hebbian rule:

$$\tilde{W}_{ij}^t = W_{ij}^{t-1} + \epsilon\left(\Psi_i^t r_j^t - \bar{\Psi}_i^{t-1}\bar{r}_j^{t-1}\right), \tag{6}$$

with a rate $\epsilon = 0.002$. $\bar{\Psi}_i^t$ and $\bar{r}_j^t$ are estimated mean firing rates of mEC unit $i$ and place unit $j$ that are adjusted at each time step of the simulation:

$$\begin{aligned}
\bar{\Psi}_i^t &= \bar{\Psi}_i^{t-1} + \eta\left(\Psi_i^t - \bar{\Psi}_i^{t-1}\right), \\
\bar{r}_j^t &= \bar{r}_j^{t-1} + \eta\left(r_j^t - \bar{r}_j^{t-1}\right),
\end{aligned} \tag{7}$$

with $\eta = 0.05$ a time averaging factor. After each learning step, the provisional $\tilde{W}_{ij}^t$ weights are normalized into unitary norm:

$$\sum_j W_{ij}^{t2} = 1. \tag{8}$$

Units that win during competitive learning (enforced by Equation 5) manage to establish strong connections with units that provide strong inputs. As learning proceeds, the units establish fields where they both receive strong inputs and, at the same time, are recovering from adaptation. The emergence of the grid map is the product of averaging over millions of time steps. It remains to be

assessed whether this mechanism we propose might also account for the formation of new grid representations in a novel environment the animal adapts to, for a sufficient time, or if, instead, it can only be applied to the developmental period.

## Grid alignment: head direction input

Two substantial extensions are represented by the introduction in the model of head direction information, through the assignment of preferred directions to mEC units, and by the presence of recurrent connections in the mEC layer beside the feed-forward set between the two layers (*Si et al., 2012*). Both these additions to the earlier version of the model are important for the grid alignment issue that we are going to study in the 3D case. With these two additional elements, the overall input to unit *j* is now:

$$h_t^i = f_{\theta_i}(\omega_t)\left( \sum_j W_{ij}^{t-1} r_j^t + \rho \sum_k W_{ik} \Psi_k^{t-\tau} \right), \qquad (9)$$

with $\rho = 0.1$ a factor setting the relative strength of feed-forward ($W_{ij}^t$) and collateral weights ($W_{ik}$), and $\tau = 25$ steps a delay in signal transmission, as discussed by *Si et al. (2012)*. The multiplicative factor $f_{\theta_i}(\omega_t)$ in *Equation 9* is a tuning function which is maximal when the current direction of the animal movement $\omega_t$ is along the preferred direction $\theta_i$ assigned to unit *i* (*Zhang, 1996*).

$$f_\theta(\omega) = c + (1-c)\exp[\nu(\cos(\theta - \omega) - 1)], \qquad (10)$$

where $c = 0.2$ and $\nu = 0.8$ are parameters determining the minimum value and the width of the cell tuning curve. Preferred head directions are randomly assigned to mEC units and they uniformly span the $4\pi$ solid angle.

## Collateral weights

The appearance of fields in the output layer of the model is fully independent of the presence of collateral connections. Instead, their basic function is to favor the appearance of a certain phase shift of the fields in the post-synaptic unit relative to the pre-synaptic one, and consequently to induce the alignment of grids, producing a common orientation in the population (*Si and Treves, 2013*). In this study we will not deal with collateral weight self-organization, only with feed-forward weight learning. For simplicity, the collateral weights are set at convenient values at the beginning of the simulations and left unchanged afterwards. We will deal with the issue of recurrent connection plasticity in future work (but see *[Si and Treves, 2013]*).

Collateral weights are set in the following way (*Kropff and Treves, 2008*): each mEC unit is temporarily assigned a preferred position, an auxiliary field at coordinates ($x_i$, $y_i$, $z_i$). The coordinates are randomly chosen among the place field centers of the input layer. These auxiliary fields are introduced only for the sake of weight definition and do not play any role in other parts of the simulations. The collateral weight between unit *i* and unit *k* is then calculated as

$$W_{ik} = \left[ f_{\theta_i}(\omega_{ik}) f_{\theta_k}(\omega_{ik}) \exp\left( -\frac{d_{ki}^2}{2\sigma_f^2} \right) - \kappa \right]^+, \qquad (11)$$

where $[*]^+$ denotes the Heaviside step function, $\kappa = 0.05$ is an inhibition factor to favor sparse weights, $f_{\theta_i}(\omega_{ik})$ is the tuning function defined above (in *Equation 10*), $\omega_{ik}$ is the direction of the line connecting the auxiliary fields of unit *i* and *k*, $\sigma_f = 0.2\,L$ defines how broad the connectivity is, and $d_{ki}$ is defined as

$$d_{ki}^2 = [x_i - (x_k + l\cos(\omega_{ik}))]^2 + [y_i - (y_k + l\cos(\omega_{ik}))]^2 + [z_i - (z_k + l\cos(\omega_{ik}))]^2, \qquad (12)$$

that is, it is the distance between the auxiliary fields with an offset $l = v \times \tau$.

The normalization on this set of connections is performed as in *Equation 8*. The definition of the weights is such that it generates strong positive interactions between cells with similar preferred head direction and activation fields appropriately shifted along the same head direction.

## Analytical model

The self-organization process we consider at the single-unit level can be described in analytical terms as an unsupervised optimization process, if one neglects the collateral interactions that are presumed to align the grids (*Si et al., 2012*). The simplified version of the model which can be analyzed mathematically is very abstract, and does not specify most of the parameters necessary to the

simulations. Nevertheless, it indicates which are the asymptotic states that should be approached by the system after having evolved for a long time.

The asymptotic states are defined in terms of a variational principle, amounting to the minimization of a cost function of the form:

$$H = H_K + H_A = = \int d\chi [\nabla \Psi(\chi)]^2 + \gamma \int d\chi \int dt' \Psi(\chi(t)) K(t - t') \Psi(\chi(t')),$$ (13)

where $\chi$ is the spatial coordinate and $\Psi$ represents the firing rate of the neuron across the environment. The functional is defined based on the hypothesis that the activity of the units reflects only two simple constraints:

1. The minimization of the variability of the maps across space, that is, a preference for smooth maps. Such smoothness is expected to stem from the smoothness of the spatial inputs and of the neuronal transfer function. This constraint is expressed in the first term of the functional, the *kinetic* one.
2. The penalization of maps that require a neuron to fire for prolonged periods of time, which is opposed by neuronal fatigue. The second term of the functional, the *adaptation* term, represents this constraint.

The parameter $\gamma$ parameterizes the relative importance of the two constraints.

The dependence of the functional on time can be eliminated by taking into account the averaging effect of a long run over many trajectories and different speeds experienced during training. We therefore substitute the time-dependent kernel $K(t - t')$ in the second term of *Equation 13* with an effective space-dependent one, $K(\chi' - \chi)$, using the average speed of the animal to fix the relationship between the two:

$$H = H_K + H_A = = \frac{1}{V} \int_\Gamma d\chi^d [\nabla \Psi(\chi)]^2 + \gamma \frac{1}{V} \int_\Gamma d\chi \Psi(\chi) \int_\Gamma d\chi' \Psi(\chi') K(|\chi' - \chi|),$$ (14)

where we have also made explicit the normalization by the area $V$ of the $d$-dimensional environment $\Gamma$.

We directly apply this expression to ask which is the favorite arrangement of the fields in a 3D volume $V$.

## Optimal packing

Unlike on the plane, where the hexagonal tiling is always the optimal one, three-dimensional space admits a multitude of equally optimal orderings of spheres. The problem of sphere packing, in a volume of 3D space, is a well-known mathematical problem, and it is known since the time of Gauss that any of these infinite optimal solutions can be described in terms of two fundamental arrangements, called face centered cubic (FCC) and hexagonal close packed (HCP), that represent the only two regular solutions to the problem. Both solutions are based on a series of layers of spheres arranged in a hexagonal pattern. These layers are stacked one upon the other with given phase differences between them (*Figure 2*). The difference between FCC and HCP lies in the sequence with which these phases appear. Given one of these layers, and taking it as a reference with positioning A, there are two possible arrangements (B and C) of the next layer, obtained with a translation of A, that puts all the spheres at the same distance from their neighbors. Any sequence of A, B and C without immediate repetitions has the same, maximum, packing score, but among all sequences there are the two regular prototypes:

- **FCC** = ABCABCABCA.
- **HCP** = ABABABABAB.

In both combinations each sphere has 12 first neighbors, and if $d$ is the diameter of a sphere (or the distance between the centers of two neighboring spheres), then the inter-layer separation is $\frac{\sqrt{6}}{3} d$. If we consider the unit cell of 13 spheres (a central sphere + 12 neighbors) in *Figure 2*, then FCC and HCP differ only regarding the position of three spheres (compare the position of the fields marked in green and red in the top-left and top-right panels in *Figure 2*). In fact, while in FCC neighbor spheres are arranged in six pairs with symmetrical positions with respect to the center, in HCP there are only three of these pairs, those on the central plane (fields marked in blue in *Figure 2*, top right).

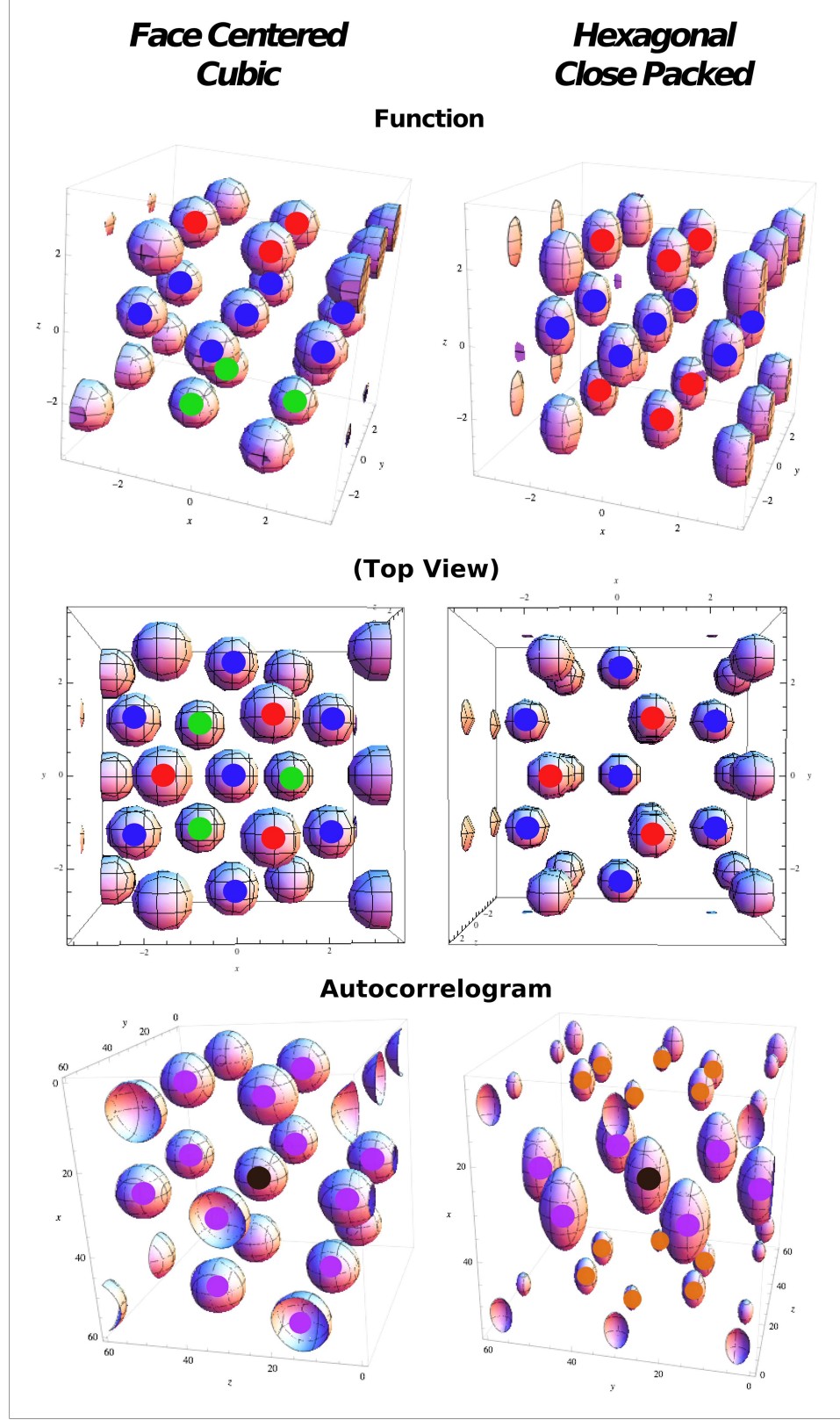

**Figure 2**. Regular optimal solutions to the sphere packing problem in 3D. Top: Functions used in cost function calculation for face centered cubic (FCC) and hexagonal close packed (HCP) structures. Color markers are used to indicate the layering in the field arrangement: FCC results from an A(red)B(blue)C(green) sequence, HCP from an A
*Figure 2. continued on next page*

*Figure 2. Continued*

(red)B(blue)A(red) sequence. Middle: Same as above but from a different viewpoint to highlight the different organization of the relative phases. Bottom: Autocorrelograms of the above functions. Color markers indicate the magnitude of the corresponding peak in the autocorrelogram. Purple: full peaks. Orange: half peaks.

Considering the three-dimensional distribution of activity $\psi(x)$ that minimizes the cost function in *Equation 14*, we then compare the relative optimality of FCC and HCP configurations. To do so we define two analytical expressions for the two arrangements of fields in terms of a combination of plane waves, as they are well suited to be treated in this formulation of the problem.

## FCC symmetry

To represent the FCC arrangement of fields we use the following expression:

$$\psi^{FCC}(r) = 1 + \frac{1}{4} \sum_{i=1}^{4} \cos(k_i \cdot r), \tag{15}$$

a combination of four plane waves (*Figure 2*, top left), with the four wave vectors $k_i$ given by the matrix:

$$k_i = \frac{2\pi}{a} \begin{pmatrix} 0 & 0 & \sqrt{3/2} \\ 2/\sqrt{3} & 0 & -1/\sqrt{6} \\ -1/\sqrt{3} & 1 & -1/\sqrt{6} \\ -1/\sqrt{3} & -1 & -1/\sqrt{6} \end{pmatrix}. \tag{16}$$

The directions of the wave vectors are equivalent to those of the center-to-vertex axes in a tetrahedron. This choice gives

$$\text{Spacing} = a. \tag{17}$$

$$\text{Normalization} < \psi^{FCC} > = 1. \tag{18}$$

$$|k_i|^2 = \frac{3}{2} \left( \frac{2\pi}{a} \right)^2. \tag{19}$$

As any power of the previous expression maintains the same symmetry properties, we will actually compute the cost function for the first few powers:

$$\psi_n^{FCC}(r) = p_n^{FCC} \left( 1 + \frac{1}{4} \sum_{i=1}^{4} \cos(k_i \cdot r) \right)^n. \tag{20}$$

Here $p_n^{FCC}$ is used to maintain the normalization.

The first term of the cost function, the spatial average $<\psi_n \nabla \psi_n>$, can then be evaluated analytically by expanding $\psi_n$ over the Fourier modes and taking into account the orthonormality relations of planar waves. This calculation is quite simple for low powers, but the number of terms increases rapidly with $n$. The resulting formulas for $n = 2$ are reported in the 'Materials and methods' section.

The second term can be similarly calculated by using the change of variable $\mathbf{q} = \mathbf{x}' - \mathbf{x}$ together with the trigonometric property:

$$\cos(\mathbf{k} \cdot \mathbf{q} + \mathbf{k} \cdot \mathbf{x} + \phi) = \cos(\mathbf{k} \cdot \mathbf{q}) \cos(\mathbf{k} \cdot \mathbf{x} + \phi) - \sin(\mathbf{k} \cdot \mathbf{q}) \sin(\mathbf{k} \cdot \mathbf{q} + \phi). \tag{21}$$

Since $\sin(\mathbf{k} \cdot \mathbf{q}) K(\mathbf{q})$ is an odd function and the integration domain is symmetrical around $\mathbf{q} = 0$, the second term in *Equation 21* does not survive the first integral in $H_A$ (*Equation 14*).

The calculations can then be performed as in the previous case after introducing the 3D Fourier transform of the adaptation kernel $K$:

$$\tilde{K}(k_i) = \int_V dq K(q) \cos(k_i \cdot q). \tag{22}$$

The adaptation kernel may take various forms, but for reasons that will become clear in the next section, here we consider a kernel in a form that makes it factorable over the spatial variables. We use a difference of radially symmetric Gaussians:

$$K(q) = K_L(q) - \rho K_S(q) = = \frac{1}{(2\pi v \tau_L)^{3/2}} \exp\left[-\frac{q^2}{(2v\tau_L)^2}\right] - \frac{\rho}{(2\pi v \tau_S)^{3/2}} \exp\left[-\frac{q^2}{(2v\tau_S)^2}\right]. \tag{23}$$

The Fourier transform of this kernel is:

$$\tilde{K}(k_i) = \tilde{K}_L(k_i) - \rho \tilde{K}_S(k_i) = = \exp\left[-\frac{1}{2}(k_i v \tau_L)^2\right] - \rho \exp\left[-\frac{1}{2}(k_i v \tau_S)^2\right]. \tag{24}$$

Again the computation of the integral, although conceptually straightforward, becomes increasingly demanding with higher values of $n$ due to the explosion in the number of terms.

## HCP symmetry

Since the hexagonal close packing does not have central symmetry, the choice of a function reproducing the arrangement of fields is less evident. We opt for:

$$\psi_n^{HCP}(r) = p_n^{HCP} \bigg( (1/2 + 1/2\cos(k_z \cdot r)) \left[1 + \frac{2}{3} \sum_i^3 \cos\left(k_{xy}^i\right) \cdot r\right]$$
$$+ (1/2 + 1/2\cos(k_z \cdot (r + \Delta z))) \left[1 + \frac{2}{3} \sum_i^3 \cos\left(k_{xy}^i\right) \cdot (r + \Delta x)\right] \bigg)^n, \tag{25}$$

where two separate wave vectors are present: $k_{xy}$ fixing the spacing on planar hexagonal layers, and $k_z$ used instead to regulate the distance between layers (*Figure 2*, top right). As in the previous case, we consider different powers of the same formula, as they all present peaks in the same configuration.

The components of $k_{xy}$ are, again

$$k_i = \frac{2\pi}{a} \begin{pmatrix} 2/\sqrt{3} & 0 \\ -1/\sqrt{3} & 1 \\ -1/\sqrt{3} & -1 \end{pmatrix}, \tag{26}$$

with $|k_{xy}|^2 = \frac{4}{3}(2\pi/a)^2$, while the $z$ component is set to $|k_z| = \frac{\sqrt{3}}{2\sqrt{2}}(2\pi/a)$ and $|k_z|^2 = \frac{3}{8}(2\pi/a)^2$. To obtain the correct HCP arrangement of fields, $\Delta x$ and $\Delta z$ should be set to:

$$\Delta x = \left(\frac{1}{\sqrt{3}}, 0\right). \tag{27}$$

$$\Delta z = \sqrt{\frac{2}{3}}. \tag{28}$$

Spacing and normalization are the same as for FCC.

The convenience of choosing a factorizing Gaussian kernel (*Equation 23*) is now evident: it allows splitting of the integrals in *Equation 14* into the $xy$ and $z$ component. Apart from this expedient, the calculations for the HCP function follow the same line of those previously described for the FCC case. A significant difference is the dependence of the HCP solution on two parameters $k_x$ and $k_z$. Therefore, while the choice of the adaptation parameters $\tau_L$ and $\tau_S$ fixes one of the two (as shown in *Figure 3*, left), one can still optimize over the ratio between the two. The value of $k_z/k_{xy}$ that should be observed in the presence of a perfect HCP pattern can be calculated as $3/32^{1/2} \approx 0.53$. In *Figure 3* (right), we plot the values obtained for different powers of our expression for the HCP arrangement. While for higher values of $n$ the value of $k_z/k_{xy}$ extracted from the optimization progressively approaches the theoretical one, interestingly the $n = 1$ case exhibits a very different behavior with an optimal $k_z/k_{xy} = 0$, independently of the value of $\gamma$. As $k_z$ represents the reciprocal of the inter-layer spacing (and also the wavelength of the activity modulation along the z-axis), this value indicates that

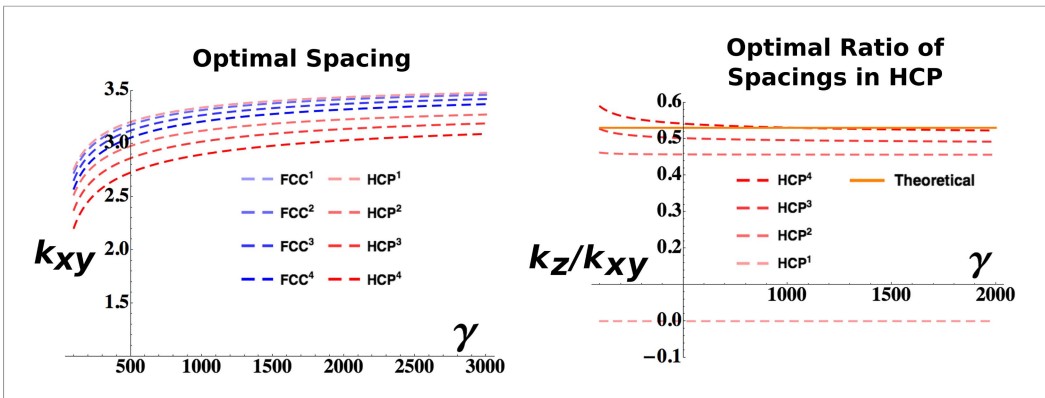

**Figure 3**. Analytical estimation of the optimal grid parameters. Results for the grid parameters resulting from the cost function minimization. Left: Optimal grid spacing for face centered cubic (FCC) and hexagonal close packed (HCP) structures and various powers of the respective expressions. Right: Optimal ratio between horizontal spacing and vertical spacing for different powers of the HCP expression. The orange continuous line indicates the theoretical value to obtain a perfect HCP arrangement. All the plots are obtained with parameters: $v\tau_L = 1$, $v\tau_S = \frac{1}{3}v\tau_L$, $\rho = 0.03$.

the minimum value of the cost function is obtained with infinite distance (and infinitely slow modulation of the activity), or equivalently with a column-like distribution of activity, with a single layer of fields extending indefinitely in the vertical direction (*Jeffery et al., 2013*). This solution is distinct from the case of a single layer of fields, with no activity above and below them, a situation that does not entail the regular, three-dimensional configurations we are interested in. In *Figure 3* we plot the values of the wave vectors obtained with the set of parameters: $v\tau_L = 1$, $v\tau_S = \frac{1}{3}v\tau_L$, $\rho = 0.03$. Changing these parameters alters the absolute values of the curves but does not affect the qualitative behavior of the solutions to the minimization of the cost function.

## Results

### Asymptotic states for individual grids

#### Self-organized grids appear to express a mixture of symmetries

In our simulations the activity of mEC units is progressively shaped over an extended period of time. Starting from an initial random arrangement of connections and a correspondingly heterogeneous distribution of activity, the combined effect of adaptation and synaptic learning leads the units to approach, after a transient reorganization of their firing fields in space, a stable configuration that is the outcome of the self-organization process. Looking at the firing configurations developed by the units in our simulations, one can notice a similarity with the theoretical solutions maximizing the packing density of spheres described above. In *Figure 4* we show two typical examples from the units emerging in the model mEC layer in two distinct simulations, each taken after about 15 million time steps of learning time. On the top row the firing rates of the units presents a blobby appearance, with equally sized, spherical fields homogeneously distributed in the volume. Although rate maps resemble those we expect from a regular tiling of the three-dimensional environment, they are not very informative about the overall organization of the fields nor about any symmetry in their spatial distribution. Computing the corresponding autocorrelograms we indeed find that the two units are rather different in this respect, as they express two distinct field configurations. One is in fact presenting an approximately FCC arrangement (*Figure 4*, left), while the other is close to an HCP one (*Figure 4*, right). Overlaying the axes of symmetry of the two ideal arrangements (green and orange lines) illustrates the differences between the two.

Thus, self-organization based on synaptic adaptation can have multiple outcomes, leading to equivalently stable solutions. Identical systems, with the same network properties and subject to the same evolution dynamics might approach different asymptotic states, just as an effect of the random initial conditions.

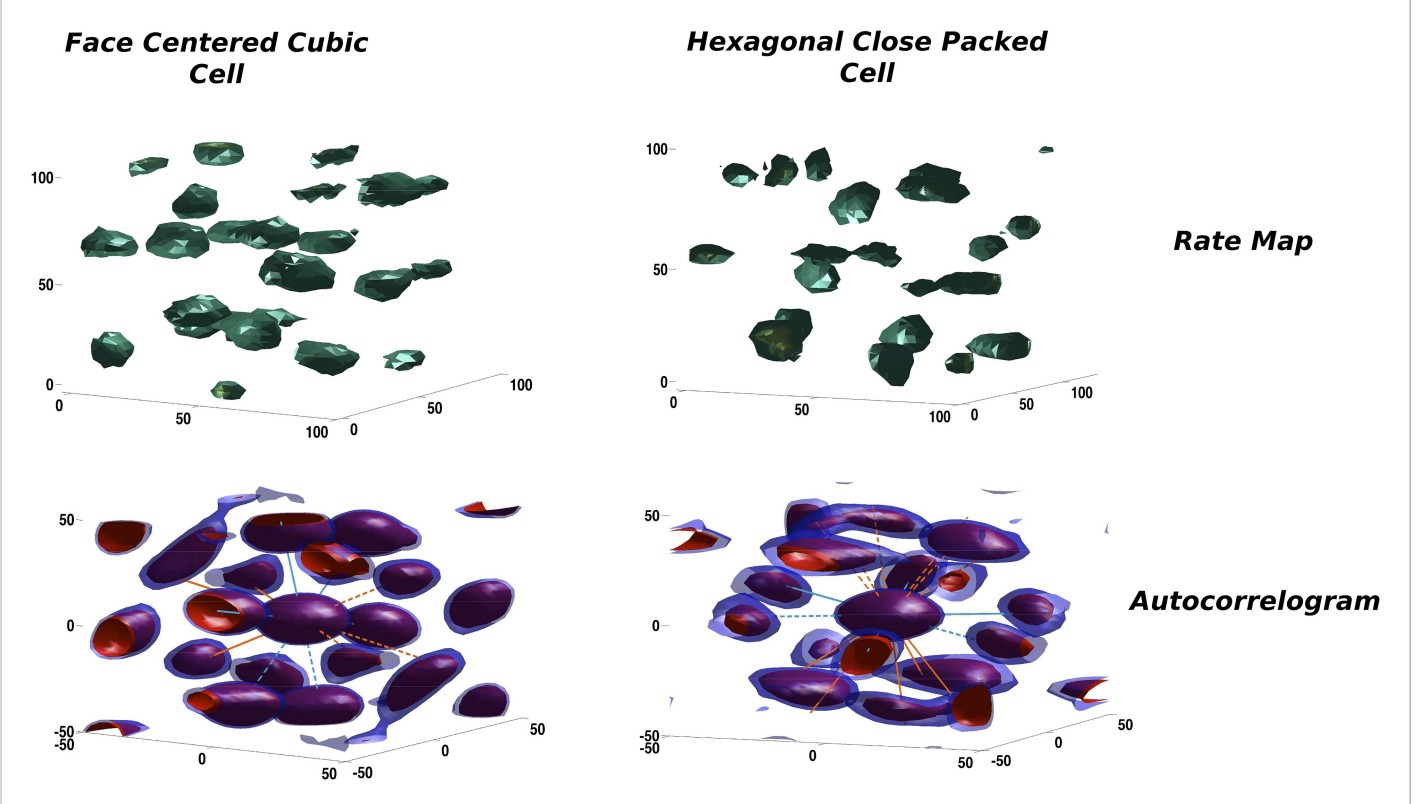

**Figure 4**. Two illustrative examples of unit activity obtained from simulations. Left: Face centered cubic (FCC)-like unit. Right: Hexagonal close packed (HCP)-like unit. Top: Rate maps. Bottom: Autocorrelograms. We plot the central portion of the autocorrelogram comprising the peaks surrounding the origin. Red and blue contours correspond to a correlation value of 0.2 and 0.25, respectively.

This fact does not necessarily imply that different solutions can coexist in the same population. The presence of interactions between units might indeed induce a global response to the initial conditions driving all the cells to develop the same symmetric properties. By making use of the measures of order described in the 'Measure of long-range order' section in 'Materials and methods' and calculating the similarity of each activity pattern either to an FCC or to an HCP, we can assess the presence within a population of mEC units of the different asymptotic arrangements. In *Figure 5* we show the distribution of the two scores, again taken after a long learning time (15 million time steps), for units all belonging to the same mEC population. The values of the two scores indicate the presence of both arrangements in the system. If we look at the scores for each unit at a given time (*Figure 5*, left), we indeed find that these are not clustered in two groups, each one expressing a homogeneous HCP or FCC arrangement, but instead they cover an entire continuum of scores between the two extremes, expressing all intermediate arrangements.

## Which is the most favorable analytical solution?

This result points to a high degree of independence of each unit and at the same time to a rather weak preference of the system for either of the regular configurations of fields. We can contrast our observations on the asymptotic states approached by the mEC units simulated numerically with the analytic evaluation of the cost function associated with the regular asymptotic states (see the 'Measure of long-range order' section in 'Materials and methods'). The calculations are based on the functional in *Equation 14*, comprising two terms, one representing the degree of smoothness of the map, the other the effects of adaptation. Its minimization (adjusting regular solutions to fit with the firing rate adaptation time scale) shows a rather complex interplay between different possible states (*Figure 5*, right). First of all, considering FCC and HCP separately, we see how, for both of them, solutions of higher power become successively favored as the value of the $\gamma$ parameter increases, that is, as the

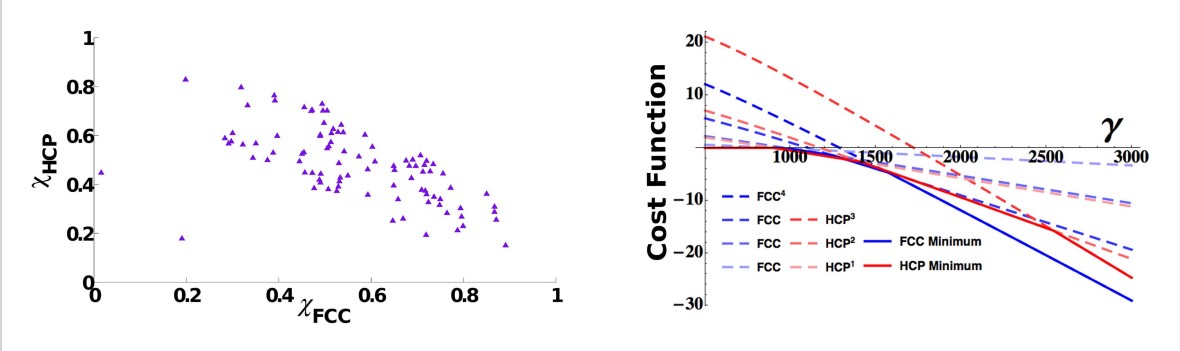

**Figure 5**. 3D grid units do not converge towards a common arrangement of fields. Left: Distribution of face centered cubic (FCC) and hexagonal close packed (HCP) scores in a population of simulated mEC units. The scores span a continuum between a pure FCC arrangement (bottom right corner) and a pure HCP one (top left corner of the figure) and are widely distributed between the two extremes. Right: Analytical cost function for various powers of FCC and HCP as a function of $\gamma$ parameter. Continuous lines indicate the minimum value within each set of solutions and are found to alternate as a global minimum in different ranges of values of $\gamma$. For small values of the parameter $\gamma$, instead, the trivial solution $\psi = 0$ is favored. The plot is obtained with: $\nu\tau_L = 1$, $\nu\tau_S = \frac{1}{3}\nu\tau_L$, $\rho = 0.03$.

weight of the adaptation component becomes increasingly dominant in the evaluation of the cost function. Therefore, the same arrangement of field positions, but with the fields becoming more and more concentrated and peaked, is selected moving rightward on the graph. In parallel, the two configurations, FCC and HCP, compete with each other, and the two are alternating as the optimal solution in different portions of the parameter space. The picture that emerges from this analysis is one of close equivalence of FCC and HCP in terms of optimality. There is no evidence of a regime in which one of the two strongly dominates the other. Instead, the features that are common to the two, like the hexagonal arrangement and the 12 first neighbors, appear to be the relevant ones for the evaluation of the cost function, with the differences appearing when considering higher order features only marginally contributing to its value and therefore generating minuscule quantitative differences between the two.

Analytical calculations deal with an abstract and simplified formulation of the properties of our network. The conclusion they suggest, however, is supported by our observations from the numerical simulation of the full model. The system does not appear to converge asymptotically to either of the two regular configurations of fields. Although neither FCC- nor HCP-symmetric solutions provide a unique description of the final arrangement of the units, the simulations produce examples of units similar to either of the two optimal packing solutions. The discrepancy between the configurations observed and the symmetric solutions, however, is not due to the emergence of mixtures of the two (like for example in a ABABCAB ordering of the layers), but rather to small deviations of the relative phases between fields of different layers from the optimal ones. Although we tested our model in environments of different size (as discussed in a following section), the model is computationally extremely demanding to run in very large environments, large enough to investigate this sort of mixed ordering. Analytical results show a substantial equivalence of FCC and HCP ordering, thus implying, in large structures, the possibility of multiple switching between the two, without altering the overall optimality of the configuration. Nevertheless, simulations show that this equivalence, even in the case of the small environments we tested (where we could usually observe three or only slightly more layers at the same time), is expressed in intermediate arrangements which are only partially symmetric. It is thus likely that introducing additional layers would just propagate this situation further without leading to the appearance of FCC and HCP mixtures.

## Time scales for the emergence of local order

Constructing either an FCC or an HCP arrangement of fields is a rather articulated endeavor that requires assembling a hierarchy of elements of increasing complexity. The three-dimensional structure described by the two arrangements implies the establishment of long-range relationships between the level of activity at distant points of the environment and involves determining the position of

a large number of fields at the same time. Both FCC and HCP are described by unitary cells of 13 fields and the difference between the two lies in the different positioning of just three of them with respect to the others. It is evident by looking at *Figure 2*, however, that this long-range order is constructed from a set of building blocks that express symmetries and regularities at a local level, involving fewer fields and a smaller set of constraints. Understanding the outcome of the self-organization of grid cells in three dimensions can be thus approached bottom-up, starting from basic features of the representation and then following the learning process up towards their combination into overarching structures.

As in the two-dimensional case, a description of the grid can start from computing the mean distance between first-neighbor fields and the mean angle formed by triplets of adjacent fields. These two measures involve, respectively, two and three fields at a time and are not informative about correlations extending beyond these boundaries.

At this level order emerges almost immediately (*Figure 6*). The mean angle among neighboring fields (calculated over all the triplets of all the cells of a simulated population) is close to $\pi/3$ from the very beginning of learning and the real effect of continuing exploration is the reduction of the variability over the course of about 4 million time steps.

A similar behavior is observed when plotting the value of the mean spacing of the grids in time (calculated from the unit autocorrelograms) (*Figure 6*, right: blue line). Also in this case, after a short transient the value stabilizes after around 3–4 million time steps. Our choice of model parameters (and specifically of the adaptation parameter) leads to a spacing of $0.55 \times L$. We run simulations in different conditions to test the sensitivity of this quantity to specific components of the model. We consider the case of having no internal connectivity in mEC, removing any interaction between different mEC units (*Figure 6*, right: red line) that are therefore developing grids independently, and the case in which rather than having a single value of the adaptation time course, common to all the units, the population expresses a range of possible values, drawn from a uniform distribution ranging from $0.85 \times b1$ to $1.2 \times b1$, where $b1 = 0.1$ is the value otherwise used (*Figure 6*, right: green line). In both cases we see that the time course of the development of a common grid spacing is not affected by the modifications of the standard model. Stabilization is obtained in the same time interval and while removing collateral connections appears to have absolutely no effect, the variability in the adaptation parameter results in a slightly different final value of the spacing ($0.5 \times L$).

These results indicate that the three-dimensional grid develops from the same ingredients of its lower dimensional equivalent. Mean spacing and mean angle are quickly fixed over the entire network almost simultaneously and are the first recognizable signs of the emergence of an ordered structure from the initial random distribution of activity. The equivalence between this process and that

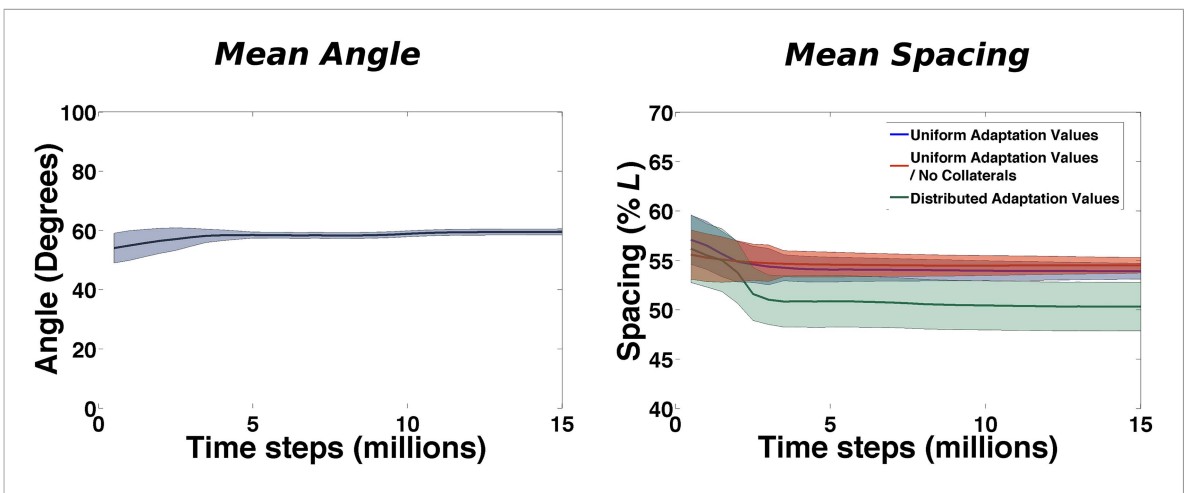

**Figure 6**. Emergence of local regularities in the arrangement of fields. Left: Mean angle formed by triplets of fields as a function of learning time (see the 'Local gridness measure' section in 'Materials and methods' for details on the measure). Right: Mean spacing between fields extracted from the autocorrelograms, for various conditions as a function of learning time.

observed in a model of two-dimensional grid development is due to the same principles driving self-organization. The presence of an additional dimension does not affect the way in which fields are initially brought by adaptation to homogeneously and regularly cover the entire space.

## Time scales for the emergence of long-range order

Using the measure described in the 'Local gridness measure' section in 'Materials and methods', we can evaluate the difference between the distribution of activity of a unit and a random arrangement of fields. Plotting the average across units of this index, which reflects the decrease of the variance in the angles between triplets, already observed in *Figure 6*, we see again (*Figure 7*, top left), that after roughly 4 million time steps the system is already arranged in a stable ordered manner, with equilateral triangles among neighboring fields that dominate the activity pattern. This ordering can be generated by the system independently of higher order symmetries, and it provides a first step for further arranging fields in more articulated structures.

A second step taken by the system is the coordination of multiple field triplets to arrange them in a hexagonal pattern. This process corresponds to the formation of a grid on the plane, but in three dimensions it involves the creation of not just one hexagon but of multiple superimposed planes each of which contains hexagonally arranged fields. To investigate the structure of the activity on the single layers, we take multiple slices of the autocorrelogram matrix. We take sections passing through the center, with different angles of azimuth and elevation. We then compute the autocorrelogram values on each of these planes, and we compare it with a hexagonal template of equidistant peaks with $\pi/3$ periodicity. The plane most resembling a hexagonal pattern according to a correlation measure (the 'best plane') is selected together with its similarity score. This method provides us with an equivalent of the traditional grid score used to judge the quality of planar grid cells. In *Figure 7* (top right) we plot the time evolution of the average over the population of this score. Starting from very low values, indicative of a still unorganized ensemble of fields, the score steadily rises to reach a value of about 0.85 (out of a maximum of 1) after 6 million time steps and then remains stable over the rest of the simulation.

The previous analysis considers each unit separately and does not provide a measure of the coordination across the population of the formation of the field layering. But the presence of collateral interactions should induce some coordination in the way these layers are arranged in different cells, that is, in the direction along which these best planes align in space. We looked for the dispersion of their orientations, calculating the angles formed by the best planes of different cells ($\beta$, *Figure 7*, middle left), and the angle between the hexagonal grid axis of cells sharing a similar best plane ($\omega$, *Figure 7*, middle right). We indeed see that collateral interaction tends to align in time the principal layer of different cells, defining a preferred orientation for the global structure of the grid. The emergence of the common alignment of the layers is slower than the formation of the layers themselves. It takes about 12 million time steps of exploration time to significantly reduce the dispersion of the angles (note that $\beta$ is slightly faster, maybe indicating a tendency to first define the layers and then arrange shifts within them).

Having hexagonal layers tiled upon each other along the same direction still leaves to each cell the degree of freedom of setting the relative phase between pairs of these layers. It is clear at this point that a proper choice of these phases would result in the reproduction of either an FCC arrangement or an HCP arrangement. This higher level in the hierarchy of order for three-dimensional grids, which when attained would provide a completely regular tiling of the volume, does not have a correspondence in the two-dimensional case. It is thus a completely new level of complexity that can be only expressed when producing grids in three dimensions. In the bottom panels of *Figure 7* we show the time course of the scores for FCC and HCP similarity (see the 'Measure of long-range order' section in 'Materials and methods'). Their values are almost unchanged over the duration of the simulations and after 30 million time steps, a time largely exceeding that necessary for the other quantities to converge, both of them are still close to 0.5, which reflects the presence in the population of a large distribution of values.

Our simulations are able to distinguish a hierarchy in the time course leading to the formation of three-dimensional grids. Different levels of complexity appear in the arrangement of fields with different speed. Fast converging quantities like the formation of equilateral triangles of fields and successively of layers of fields with hexagonal symmetry appear first, framing the activity of single

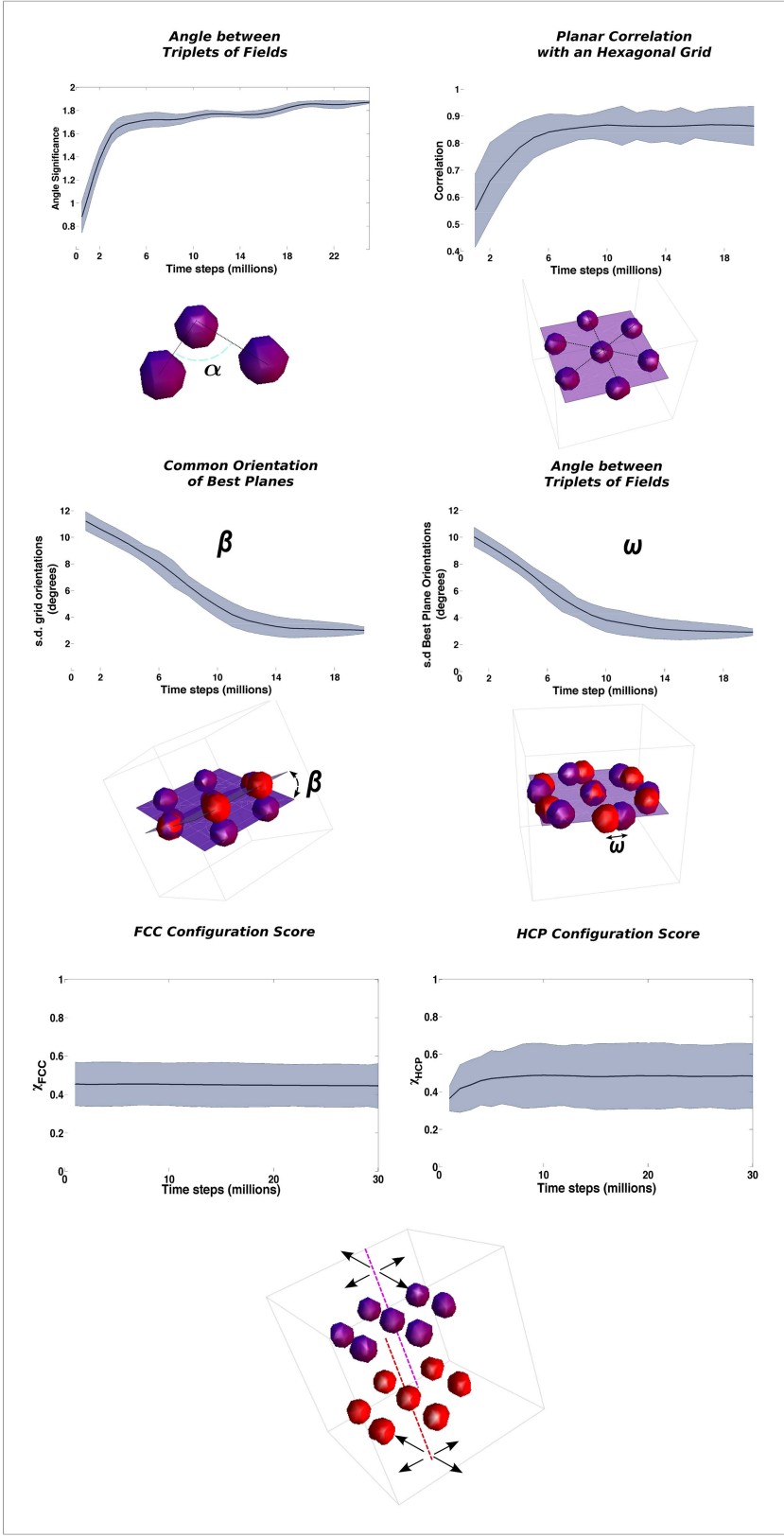

**Figure 7**. Three distinct time courses for the emergence of long-range ordering. The panels present the time evolution of the measure of different symmetries in the arrangement of fields. Top row: Fast convergence of neighboring triplets of fields towards equilateral triangles (left) and of group of fields into planes with a hexagonal

*Figure 7. continued on next page*

*Figure 7. Continued*

arrangement (right). Middle row: Slow convergence of the different units in the population towards a common orientation of the layers of fields. Left: Angle between the principal plane expressed by the various units. Right: Angle between fields arranged over mutually aligned planes. Bottom row: No convergence of global, inter-planar order. The measures of similarity with face centered cubic (FCC) and hexagonal close packed (HCP) ordering do not evolve with the extension of the learning time and remain close to intermediate values indicating an even distribution of the values over the population of grid cells. See 'Materials and methods' for details on the measures.

units in the network. These initial structures are then modified on a slower time scale to obtain a global coordination among cells. Planes of fields are rotated to align them across the population, generating a common tiling of the fields of different cells that conserve their unique spatial phase. This global ordering is only partial though, as the phases of the units are only partially overlapping with those necessary to reproduce a perfectly regular tiling of the volume (either with an FCC or an HCP). If we exclude the very initial phase of network dynamics, self-organization does not appear to affect these aspects of network activity that therefore remain loosely determined even after a very extended period of time.

## Volume dependence of the time scales

The critical question is how would these timescales scale up with the size of the environment. At least for the most rapid self-organizing processes, their very nature, dependent on plasticity in the feed-forward connections, would appear at first sight to require the pairing of each activity field of each unit to the specific configuration of sensory inputs which impinge at the same time on the feed-forward connections, therefore implying times for the formation of the grid that scale up with the number of fields in the volume. A volume of linear size $L$ includes roughly $N_3 = \sqrt{2}(L/a)^3$ fields of spacing $a$ in either an FCC or HCP arrangement (and $6\sqrt{2}(L/a)^3$ trajectories connecting neighboring fields). A square of side $L$ on a plane would include roughly $N_2 = (2/\sqrt{3})(L/a)^2$ fields. The emergence of equilateral triangles in 3D appears to require roughly a factor 2 more time than in 2D (*Si et al., 2012*), in approximate agreement with the factor $N_3/N_2 = \sqrt{(3/2)}(L/a){\approx}1.2 \times 1.8{\approx}2$ coming from the above argument. Note that if that were correct, equilateral triangles in an environment roughly four times as large, as can be argued to be the one used in the bat experiments in the Ulanovsky laboratory (*Ginosar et al., 2014*), would emerge in roughly 40–50 hr of continuous flight. Although the developmental maturation of the bat encompasses longer cumulative flying hours, what is likely relevant for structuring the feed-forward connections is time spent flying in the environment of the actual experiment. Not only can the mechanisms leading to grid-like activity only unfold while navigating and not during rest periods, they also appear to require, in our model, the exact configuration of input activity at each location in the environment. Therefore this constraint is likely to put the time scale of grid cell formation well above the feasible time duration of the experiment.

In fact, however, we find that the time for self-organization lengthens only a little, and clearly sublinearly, with the volume flown by the virtual bat. Given the multiple sub-processes involved in the self-organization of the grid units, we focus on a summary measure, derived from the analytical model: the cost function (*Equation 14*). Each of the two terms of the cost function, the kinetic and the adaptation kernel, can be calculated for each model grid unit at each time step of the simulation, and average values can be extracted and fit, for example, with sums of exponential functions. What cannot be calculated from the simulations themselves is the value of the $\gamma$ factor that, in the cost function, would determine the weight of the adaptation kernel with respect to the kinetic term.

We find that the population-averaged data points for both terms can be well fit by a sum of two exponentials, plus a constant (*Figure 8*, inset) and with the same time parameter for the first exponential in each term:

$$H_K(t) \simeq A_v \exp\left(-t/\tau_v^S\right) + B_v \exp\left(-t/\tau_v^L\right) + K. \tag{29}$$

$$H_A(t) \simeq C_v \exp\left(-t/\tau_v^S\right) - D_v \exp\left(-t/\tau^M\right) + E_v. \tag{30}$$

with $A$, $B$, $C$, $D$, $E$ volume-dependent positive fit parameters, and $\tau^{S,M,L}$ short, medium and long relaxation time scales ($K$ turns out not to depend on the volume; nor, it seems, does $\tau^M$).

The short term relaxation is therefore a joint decrease of both terms, while later the adaptation term rises, whereas the kinetic term continues to decrease. An empirical ansatz can be defined for $\gamma$ as the largest value that still keeps the sum $H_K(t) + \gamma H_A(t)$ monotonically decreasing. With this ansatz, we plot the estimate of the cost function (without the $E_v$ term, for clarity) for varying volume sizes, where we have multiplied either one or two of the three linear dimensions by either 1.2 or 1.4, obtaining volumes larger than the standard one by factors 1.2, 1.4, 1.44, 1.68 and 1.96. We can see from *Figure 8* that the relaxation of this estimated cost function is mainly determined by the most rapid exponential terms, and is virtually complete by 3–4 million time steps, with a limited volume dependence. Consequently, apart from the slight prolongation of the initial transient, the time evolution of our measure for volumes of different size appears to be quite similar.

These results suggest that the time required for the complex dynamics of grid development depends only weakly on the number of fields that have to be arranged in a orderly manner in the volume. The initial relaxation, which accomplishes most of the rearrangement and probably centers on adjusting the angles between triplets of fields (cp. *Figure 7*, top left and *Figure 8*), occurs in a time that increases sublinearly with system size. This is followed by some bouncing back of the adaptation term, which may have to do with the finer adjustment of most field distances, leading to planar hexagonal grids (see *Figures 6* and *7*, top right), with a time constant which can be taken to be independent of the volume. It is protracted later by continued but minor smoothing of the fields, now in place on the best planes, concurrent, if collateral interactions are included, with the adjustment of the planes with respect to each other (*Figure 7*, middle), which extends over longer times. We do note that we have observed considerable variability in the degree of smoothness of the individual fields obtained at the end of the simulation, with a tendency for the larger volumes to require longer time and end up with rougher fields. Given the variability from simulation to simulation, however, it remains to be determined whether this trend is robust and whether it points at a significant bifurcation in trajectories of grid development.

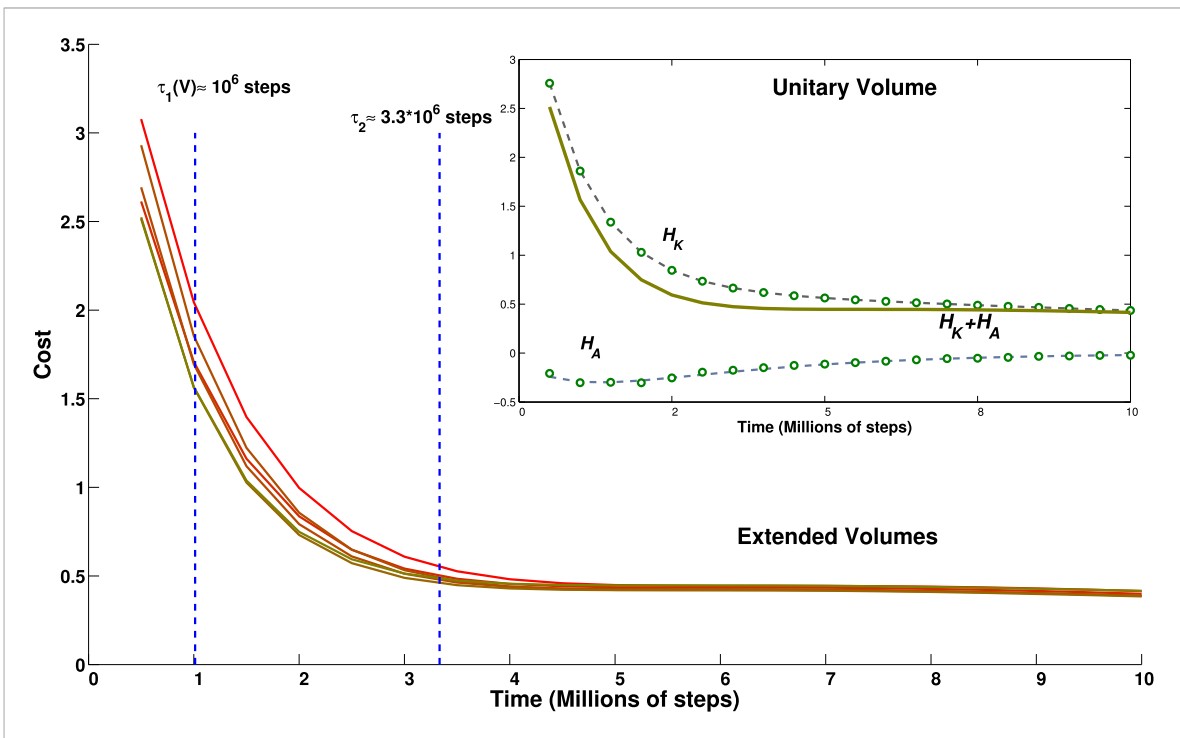

**Figure 8**. Effect of environment volume on grid developmental time. Temporal evolution of the cost function calculated for environments of different size. Lines from green to red correspond to environments of increasing size: 1 (green), 1.2, 1.4, 1.44, 1.68 and 1.96 (red) times the basic volume, respectively. With the choice of parameters reported in the discussion, these volumes would range from a 15.625 m³ room to a 30.625 m³ room. The inset shows the breakdown of the contribution to the total cost of the kinetic part and of the adaptation part for the cubic environment of size 2.5 × 2.5 × 2.5 m. Dots correspond to data points, lines to a fit. The constant of the kernel term, which varies with the volume, is not included for clarity.

## Discussion

How are these results relevant to predict the grid configuration expressed in 3D, and that can be tested in a flying bat? Our model points towards a hierarchy of timescales, associated with the emergence of periodical spatial activity of increasing complexity. To establish a relation between our results and a real bat, it is necessary to specify the actual values of the temporal and spatial parameters of our model, to obtain a time scale for the development of the grids that we can then compare with experimental findings.

If we take time steps of size $\Delta t = 10\ ms$ and an average bat velocity of $v = 1\ m/s$, the small environment used in most of our simulations will correspond to a cubic room of size $L = 2.5\ m$. Then, with this choice of parameters, grids are formed with a field spacing of $2.5\ m \times 0.55 \approx 1.4\ m$ and an interlayer distance of $2.5m \times 0.55 \times \frac{\sqrt{6}}{3} \approx 1.1m$. The time scale of grid formation can be calculated considering that 1 million simulation time steps correspond to 10,000 s or nearly 3 hr. Our model then predicts, in an environment the size of ours, the presence of (i) triplets of fields forming roughly equilateral triangles in $\approx$10–12 hr of continuous flight, (ii) hexagons in $\approx$15–18 hr, and finally (iii) different units that achieve a common orientation after $\approx$30–35 hr. These time scales do not seem very different from those predicted by the same model for the development of grids in a two-dimensional environment of similar linear size (relative to the grid spacing). Figure 6 in *Si et al. (2012)* indicates a time scale of about 20,000 s, or 5–6 hr, for the development of gridness in 2D. At the same time, this moderate increase in grid formation time might make it comparable to the flight time available for spatial learning during bat experiments. In these conditions, even the weak, sub-linear dependence of time scales with volume, that we do observe, may be sufficient to determine a switch between the possibility of forming regular structures and leaving them beyond reach.

A regular tiling of the environment (either in the form of an FCC or of an HCP lattice) is a different story, even though it would be the optimal arrangement from an information-theoretic perspective (*Mathis et al., 2014*). The total simulation time would correspond to a maximum of $\approx$80–90 hr of continuous bat flight in the $(2.5\ m)^3$ volume. This time is just a lower bound for the time necessary to form a regular tiling of the environment, and likely a loose one, as our simulations do not seem to be converging towards one of them.

These considerations suggest that bats may form a partially regular 3D tiling of the environment at most once, and then possibly only if constrained to fly for a prolonged time in a rather small cage, while a completely regular, crystalline tiling of space seems to be hardly in the range of time available to real bats.

In conclusion, the presence of an additional dimension does not seem to preclude the appearance of some orderly arrangement of the fields in mEC units of bats. Nevertheless, this order might express only a partial set of the full spectrum of potential three-dimensional symmetry properties. It might be still sufficient to distinguish the activity of these cells from a random multi-peaked pattern, but it would place it at a substantial distance from a perfectly regular pattern, too. In our model this distance varies across a population of cells: some of them show only small deviations from perfect symmetry. We thus cannot exclude the possibility of finding some of these extreme cases in real animals. At the same time, the vast majority of simulated grid cells are very far from a perfectly regular arrangement and while the number of units actually found in the tails of the distribution of scores might be strongly dependent on some specific factors of the development of grid cells, the bulk of grid-like but imperfect cells can be regarded as a robust aspect of our model, and might extend to very different situations of 3D grid cell development, possibly including other species experiencing three-dimensional navigation.

## Materials and methods

### Local gridness measure

The triangular tile is the minimal structure associated with regular volume tessellations. The two properties defining any regular triangle are the length of the side and the internal angle. Therefore, to characterize the local structure of the grid pattern in an individual unit we extract these two properties from the spikes it produces. Firstly, from our three-dimensional rate maps we generate a representative number of spike pairs through a Poisson process to construct the distribution of distances. Typically, this distribution is highly multi-peaked, where the first peak corresponds to

distances between intra-field spikes, the second peak between spikes belonging to neighboring fields, and subsequent peaks between spikes in non-adjacent fields. Since the length of the side of the tiling triangle in a regular pattern would correspond to the location of the second peak, we define a range of distances around this peak as a filter condition to declare spikes belonging to neighboring fields. The limits of this range were defined by the surrounding troughs, if they exist, or fixed to 0.5 $d$ and 1.4 $d$, if they do not, where $d$ is the distance corresponding to the second peak, declared as the grid distance of the unit. As a control condition, we generate a distribution of pseudo-spikes from reshuffling spike–cell combinations and randomly reassigning spikes to different units, thus removing the field structure of the activity of each cell. Therefore, distances between pseudo-spikes are unimodally distributed. Secondly, triplets of spikes were putatively classified as belonging to neighboring fields based on distance filtering in the previous range, and the three internal angles determined. These three angles were pooled together and accumulated in an overall angular distribution. The distribution of angles so obtained for the spiking activity and the control condition were different and their ratio was used to characterize the angle subtended in the triangular pattern. Typically (in the asymptotical state), this ratio was unimodal and distributed asymmetrically around a peak. We defined the characteristic angle as the median of the above-chance distribution (ratio values above unity indicate an above-chance condition or, in other words, angles more frequently obtained than randomly) and the significance of the angle as the maximum of the ratio distribution.

## Measure of long-range order

FCC and HCP differences in the configuration of fields generate distinct symmetry properties for the two arrangements. These symmetries are reflected in the autocorrelograms that can be extracted from them. In the same way as the autocorrelogram of a hexagonal grid is a hexagonal grid, calculating the autocorrelogram of FCC (using function 20) just reproduces the same configuration (*Figure 2*, bottom left) of fields, with six symmetric pairs of equivalent peaks surrounding the central one. Indeed the symmetries of the structure are such that one can find four planes passing through the origin which contain peaks arranged in a hexagonal way; these planes form angles of 72° and are all equivalent. The case is different for HCP, where the central symmetry is missing. In this case the autocorrelogram extracted from *Equation 25* does not reproduce the original form of the function. In *Figure 2*, bottom right, one can see that the autocorrelogram presents nine pairs of peaks around the central one. But in this case these peaks are not all the same. The HCP structure is again periodical for translations along a plane, generating six peaks of height 1, like those of FCC (*Figure 2*, purple peaks). As the structure is translated out of this preferred plane, the ABAB arrangement of the HCP layers is such that there are no translations that reproduce the exact same configuration of fields, in the autocorrelogram. The six peaks above the central one (*Figure 2*, orange peaks) are indeed half-peaks, corresponding to an overlap of only half (six out of 12) of the peaks of the basic unit. Therefore, although one can identify seven planes with hexagonally arranged peaks on this autocorrelogram, they are not all equivalent to those in FCC. Only one of them contains all the peaks of height 1 and forms an angle of 72° with the other six, which include half-peaks and form an angle of 56° between them. We can then use different measures to quantify the degree of similarity of a unit activity to the FCC and HCP prototypical field arrangements. One measure is based on the autocorrelogram. From this, we first identify the best plane, the one which yields the highest grid score, measured here as the value of the planar autocorrelogram at the origin, that is, the planar autocorrelation over all the slices passing through the origin. Once the best plane has been identified, we use the fact that the FCC has three more planes with hexagonal symmetry, at ~72° from the best plane and between one another. HCP instead has six of them, again at ~72° from the best plane, but at ~56° between them. We then take the slice scores, that is, the planar autocorrelation values on any one slice. We take all the slices at an angle of ~72° from the best plane and sum the scores of the best triplet of slices with ~72° of separation ($\zeta_{2-4}$). We then exclude them and take a second triplet of slices again with a ~72° distance from one another and a distance of ~56° from the first triplet ($\zeta_{5-7}$). These two numbers tell us about the number of different planes with hexagonal symmetry that can be built from our autocorrelograms. Both scores run from −3 to 3, as they are the sum of three correlations. We expect $\zeta_{2-4}$ to be high for both FCC and HCP arrangements, and its value should be considered as an indicator of the general quality of the grid. $\zeta_{5-7}$ instead should be high only for those grids presenting an HCP type of arrangement, but again its value might be affected by the quality of the grid. We thus define a score for the degree of FCC similarity as:

$$\chi_{FCC} = (\zeta_{2-4} - \zeta_{5-7})/\zeta_{2-4}, \tag{31}$$

that should be close to 1 in the presence of FCC, and to 0 in the HCP case.

On the other hand, HCP is characterized by the repetition of the same field positions every two layers, while FCC has a periodicity of three layers. Then another way to characterize the grids is to look for similarities between layers. Since the best plane we calculated indicates the direction of stacking of layers in the HCP (along its normal vector), we can go back to the firing rate map, take slices along this direction (that is, slices with the same best plane orientation) and calculate the correlation between planes separated by a two-layer distance ($2 \times \lambda_z$):

$$\chi_{HCP} = \rho_{auto}(2 \times \lambda_z). \tag{32}$$

Contrary to the previous score, this one should be close to 1 when HCP is expressed, and to 0 when FCC is.

## Cost function expression for $n = 2$

Here we give an example of the expression for the cost function obtained for n = 2.

$$H(\psi_2^{FCC}) = \frac{71 \times k^2}{162} + \gamma \times \left( 1 \Big/ 648 \times \left( 256\tilde{K}(k) + \tilde{K}(2k) + 6 \times \left( \tilde{K}\left(2\sqrt{2/3}k\right) + \tilde{K}\left((2k)\Big/\sqrt{3}\right) \right) \right) \right), \tag{33}$$

$$
\begin{aligned}
H(\psi_2^{HCP}) = {} & \frac{3045 \times k_{xy}^2 + 1881 \times k_z^2}{2601} \\
& + \gamma \times \Bigg( \frac{1}{3468} \times \Big( 54\tilde{K}(2k_z) + 1160\tilde{K}(k_{xy}) + 1920\tilde{K}[k_{xy} + k_z] \\
& + 36 \times \tilde{K}(k_{xy} + 2k_z] + 2 \times \tilde{K}(2k_{xy}) + 48 \times \tilde{K}(2k_{xy} + k_z) \\
& + 9 \times \tilde{K}(2k_{xy} + 2k_z) + 200 \times \tilde{K}\left(\sqrt{3}k_{xy}\right) + 36 \times \tilde{K}\left(\sqrt{3}k_{xy} + 2k_z\right) \Big) \Bigg),
\end{aligned}
\tag{34}
$$

where $k$ is the only parameter for spacing in FCC and $k_{xy}$, $k_z$ are the two spacings of HCP, along the horizontal and vertical plane, respectively.

## Acknowledgements

This work partially was supported by EU FET grant GRIDMAP. Discussions with the GRIDMAP collaboration, with its reviewers, with Andreas Herz and particularly with Nachum Ulanovsky and members of his laboratory are gratefully acknowledged.

## Additional information

### Funding

| Funder | Grant reference | Author |
| --- | --- | --- |
| European Commission (EC) | FET grant GRIDMAP | Federico Stella |

The funder had no role in study design, data collection and interpretation, or the decision to submit the work for publication.

### Author contributions

FS, Conception and design, Acquisition of data, Analysis and interpretation of data, Drafting or revising the article; AT, Conception and design, Analysis and interpretation of data, Drafting or revising the article

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
