## [Decision Letter]

Thank you for sending your work entitled “The Self-Organization of Grid Cells in 3D” for consideration at *eLife*. Your article has been favorably evaluated by Eve Marder (Senior editor), a Mark Goldman (guest Reviewing editor), and two reviewers.

The Reviewing editor and the reviewers discussed their comments before we reached this decision, and the Reviewing editor has assembled the following comments to help you prepare a revised submission:

In the interesting manuscript “The self-organization of grid cells in 3D”, Stella and Treves provide the first predictions for the development through ontogeny of 3D firing patterns in 3D grid cells. The authors pursue a two-pronged approach in which they simulate a set of weekly selective 3D place cells with feedforward plastic synapses onto a population of entering grid cells and demonstrate the formation of 3D grid fields, with the grid spacing being controlled by the time-constant of firing-rate adaptation. Additionally they employ an effective field approach to analytically predict low-energy configurations of the population firing rate maps.

Surprisingly, their model predicts that an adult network of 3D grid cells should comprise of a heterogeneous mix of both perfect (lattice like) 3D grids but also much-distorted grids. They show that, as the network is learning, already at the very beginning there appear triplets of spatial blobs of activity that are angled at 60-degrees. Only much later, with extensive training, do these triplets “crystallize”, and the network gradually develops optimal-packing structures such as the hexagonal close pack (HCP) lattice or face centered cubic (FCC) lattice - as well as neurons which exhibit a mixture of HCP/FCC. This study is very interesting in that it provides direct predictions on how and why 3D grid cells should develop during ontogeny; there is very little data on this and so it is quite interesting that the authors have managed to make headway on this topic through careful theoretical analysis, examining the space of possibilities ahead of experiments. The model also explains how and why a high heterogeneity of 3D grid cells is expected - from cells exhibiting only triplets of activity-blobs angled at 60-degrees (but without any large-scale lattice organization), through random mixtures of HCP and FCC lattices, and ending with perfectly crystallized HCP and FCC lattices. This study will be of great interest to many neuroscientists, system biologists and physicists.

Comments:

1) In the Abstract (and in many other places in the paper): The focus seems to be on predictions for bat experiments. Why do the authors single out the bat as the only relevant animal model for 3D grids? The predictions in the paper are very general, and may hold also for monkeys, dolphins, cats, or humans that also move through 3D space. The focus on bats is too narrow, and should be broadened throughout.

2) In the subsection “The network”, the authors mention that sigma_p = 0.05L. Does changing the sigma_p of the place cells affect any of the final properties of the grids, and/or their developmental time course?

3) In the end of the subsection headed “HCP symmetry”, it is unclear why the authors write that small k_z_ is equivalent to columns spreading over the z-axis (height) entire room. An alternative possibility is having only one layer of *spherical* blobs in the XY plane, without any additional layers repeating in the z-dimension. Are these options equivalent in terms of the minimization of the cost function?

4) Results section, subsection headed “Which is the most favorable analytical solution?”: The authors focus on HCP and FCC, but it is unclear here to which degree do they see also other solutions, besides HCP and FCC, e.g. random order of layers: ABCABABACBCBA… which is an arrangement that pack 3D space just as well as FCC or HCP but does not have any large-scale structure along the z-axis. Did you observe such neurons in your simulations?

5) In the Discussion: The impression one gets from the end of the Discussion is that almost no cells at all are expected to exhibit perfect FCC or HCP - but in fact, according to Figure 4 (left), it seems that at least some of the grid cells are actually expected to develop a perfect FCC or a perfect HCP; is this correct? These cells might be a minority, but at least some are expected to develop FCC or FCP. So I think it's worthwhile to write it here explicitly, because right now the discussions seems to suggest the opposite.

6) The model does not allow for plasticity between grid cells. The spatially localized structure for grid cell firing is essentially built in by hand through hand tuning, ahead of time, the lateral synapses. Note that this is only a suggestion for a future avenue of research and does not need to be addressed for the manuscript to be deemed acceptable. The progress made on addressing this difficult theoretical problem is sufficient for publication already.

7) The Results section could be written to be in a somewhat more accessible form for the general, less mathematical reader.

---

## [Author Response]

*1) In the Abstract (and in many other places in the paper): The focus seems to be on predictions for bat experiments. Why do the authors single out the bat as the only relevant animal model for 3D grids? The predictions in the paper are very general, and may hold also for monkeys, dolphins, cats, or humans that also move through 3D space. The focus on bats is too narrow, and should be broadened throughout*.

We agree, with the qualification that the prevalence of an allocentric coding of the animal’s own position in space has yet to be established for most mammalian orders. We would not want our model to be seen as taking a stand, for example, on the controversial issue of place vs. spatial view codes in monkeys. To avoid that, while still pointing at the potential generality of the conclusions, we have added, in the Abstract, “… bats, or perhaps dolphins…”, and, as a second sentence of the Introduction, “how does it code for space extending in three dimensions?”

Still in the first paragraph, we have expanded a sentence to read: “… and it provides an indication possibly valid also for other animals living and moving extensively in three dimensions, like for example dolphins, monkeys and even nonmammalian species”. And, in the second: “the form that grid cells will exhibit in higher dimensionality (currently tested in flying bats; Jeffery, 2013) is still not clear.” We have also concluded the Introduction with the sentence: “We use bats as our reference, as it is the species currently available for experiments during roughly homogeneous navigation along the three dimensions of physical space.”

In the Discussion, instead of just asking “expressed by a flying bat?” we now ask “expressed in 3D, and that can be tested in a flying bat?” and see below (point 5) for the sentence inserted at the very end of the Discussion.

*2) In the subsection “The network”, the authors mention that sigma_p = 0.05L. Does changing the sigma_p of the place cells affect any of the final properties of the grids, and/or their developmental time course?*

That paragraph has now been extended to clarify this point:

“Place field centers are homogeneously distributed in the volume (consistently with the experimental data presented in Yartsev, 2013). […] the properties of the developing grid fields depend on the time scale of adaptation and not on the size of the place fields.”

*3) In the end of the subsection “HCP symmetry”: It is unclear why the authors write that small k*_*z*_
*is equivalent to columns spreading over the z-axis (height) entire room. An alternative possibility is having only one layer of* spherical *blobs in the XY plane*, *without any additional layers repeating in the z-dimension. Are these options equivalent in terms of the minimization of the cost function?*

We have hopefully clarified this point by expanding the relevant paragraph:

“… of the inter-layer spacing (and also the wavelength of the activity modulation along the zaxis), this value […] with no activity above and below them, a situation that does not entail the regular, three dimensional configurations we are interested in.”

*4) Results section, subsection headed “Which is the most favorable analytical solution?”: The authors focus on HCP and FCC, but it is unclear here to which degree do they see also other solutions, besides HCP and FCC, e.g. random order of layers: ABCABABACBCBA… which is an arrangement that pack 3D space just as well as FCC or HCP but does not have any large-scale structure along the z-axis. Did you observe such neurons in your simulations?*

We have added this discussion at the end of section A of the Results, on the most favourable analytical solution:

“The discrepancy between the configurations observed and the symmetric solutions […] additional layers would just propagate further this situation without leading to the appearance of FCC and HCP mixtures.”

*5) In the Discussion: The impression one gets from the end of the Discussion is that almost no cells at all are expected to exhibit perfect FCC or HCP - but in fact, according to*
Figure 4
*(left), it seems that at least some of the grid cells are actually expected to develop a perfect FCC or a perfect HCP; is this correct? These cells might be a minority, but at least some are expected to develop FCC or FCP. So I think it's worthwhile to write it here explicitly, because right now the discussions seems to suggest the opposite*.

We have added a passage to clarify this point at the end of the Discussion:

“In our model this distance varies across a population of cells […] possibly including other species experiencing three-dimensional navigation.

*6) The model does not allow for plasticity between grid cells. The spatially localized structure for grid cell firing is essentially built in by hand through hand tuning, ahead of time, the lateral synapses. Note that this is only a suggestion for a future avenue of research and does not need to be addressed for the manuscript to be deemed acceptable. The progress made on addressing this difficult theoretical problem is sufficient for publication already*.

In the section on “Collateral Weights”, we have now clarified in the very beginning that “the appearance of fields in the output layer of the model is fully independent of the presence of collateral connections. Instead, their basic function…”

7) The Results section could be written to be in a somewhat more accessible form for the general, less mathematical reader.

We have made some adjustments to the Results and Methods section, for example in the first paragraph of the Methods, we have added for clarity the sentence: “The path the animal performs is generated as a correlated random walk in which the direction of movement at any time step depends on the previous one”.

We have also added some clarifications in the “Volume Dependence” subsection of the Results, where the issue of dimensional scaling is discussed.

We have introduced a new figure (Figure 1) to provide a pictorial explanation of the model and of the main idea conveyed in the Results section: the model does indeed produce three-dimensional grids but this process requires an extensive amount of time. We also extended Figure 2 to include an additional visualization of the FCC and HCP arrangements of fields from a different point of view. The panels in the second row now highlight the different tiling of the layers in the two structures.
